# Cross-ViT based benign and malignant classification of pulmonary nodules

**Qinfang Zhu**[1]\*, **Liangyan Fei**[2]

**1** Geriatric Hospital Affiliated to Wuhan University of Science and Technology, Wuhan, Hubei, China, **2** Key Laboratory of Metallurgical Equipment and Control Technology, Ministry of Education, Wuhan University of Science and Technology, Wuhan, Hubei, China

\* 2883040@wust.edu.cn

**Data Availability Statement:** The data supporting the findings in this paper are available from LIDC-IDRI database(https://www.cancerimagingarchive.net/collection/lidc-idri).

## Abstract

The benign and malignant discrimination of pulmonary nodules plays a very important role in diagnosing the extent of lung cancer lesions. There are many methods using Convolutional neural network (CNN) for benign and malignant classification of pulmonary nodules, but traditional CNN models focus more on the local features of pulmonary nodules and lack the extraction of global features of pulmonary nodules. To solve this problem, a Cross fusion attention ViT (Cross-ViT) network that fuses local features extracted by CNN and global features extracted by Transformer is proposed. The network first extracts different features independently through two branches and then performs feature fusion through the Cross fusion attention module. Cross-ViT can effectively capture and process both local and global information of lung nodules, which improves the accuracy of classifying the benign and malignant nature of pulmonary nodules. Experimental validation was performed on the LUNA16 dataset, and the accuracy, precision, recall and F1 score reached 91.04%, 91.42%, 92.45% and 91.92%, respectively, and the accuracy, precision, recall and F1 score with SENet as CNN branch reached 92.43%, 94.27%, 91.68% and 92.96%, respectively. The results show that the accuracy, precision, recall and F1 score of the proposed method are 0.3%, 0.11%, 4.52% and 3.03% higher than those of the average optimal method, respectively, and the performance of Cross-ViT network for benign and malignant classification is better than most classification methods.

## Introduction

Lung cancer is a multifactorial, granulomatous disease with inconspicuous early symptoms that usually manifests as lung nodules. The International Agency for Research on Cancer has compiled cancer data from 185 countries around the world, showing that the incidence rate of lung cancer is 11.4%, which is the second highest cancer incidence rate, and the mortality rate of lung cancer is 18% of all deaths due to cancer, which is the first place [1]. The degree of lung nodule pathology (benign or malignant) is of vital importance to the physician in confirming the diagnosis of the disease. If the benign or malignant nature of the lung nodule was determined, early interventions could be made, more targeted treatment plans could be developed, and survival rates could be improved [2].

**Funding:** The author(s) received no specific funding for this work.

Pulmonary nodules are round or irregularly shaped lesions that proliferate in the lungs and usually appear as densely shaded, well-defined or ill-defined masses on computed tomography (CT) images of the lungs [3]. Clinically, physicians use lung CT images to observe the shape and size of lung nodules. Depending on the thickness of CT slices, there are as many as dozens to hundreds of slices in one CT scan. A large amount of CT data needs to be interpreted by the doctor's professional knowledge, but this manual detection is inefficient, and the results are dependent on the doctor's personal level of practice, which is prone to omissions and misdetections leading to delaying in the patient's condition. At this point, a Computer Aided Diagnosis (CAD) system that can efficiently diagnose lung nodules is needed to assist doctors in their diagnosis, which not only greatly reduces the work pressure of doctors but also reduces leakage and misdiagnosis due to doctors' inexperience.

As early as 1980s, many researchers developed CAD programs focused on the detection and identification of pulmonary nodules in CT scans [4]. Traditional machine learning methods dealing with pulmonary nodule classification first extract information about the texture, shape and edges of the nodule, and subsequently train classifiers to make classification predictions. Wu et al. [5] developed a new clustered random forest classification algorithm that combines clustering with RF to distinguish benign and malignant nodules by class decomposition and adjusting weights. Han et al. [6] compared three 2D texture features of pulmonary nodules on CT images and found that Haralick features were more favorable for classification of pulmonary nodules as benign or malignant, and finally the features were classified using support vector machines. Farahani et al. [7] proposed a computer-aided classification method for lung CT images integrating three classifiers. Morphological features of roundness, denseness, ellipticity and eccentricity of pulmonary nodules were used in the classification process in a manner decided by each classifier itself and finally the classification results were output by voting using majority voting method. Traditional machine learning classification methods have shown significant results in the task of pulmonary nodule classification, but they rely too much on manual feature selection and are unable to extract deep-level features of pulmonary nodules.

Later, deep learning techniques were rapidly developed and widely used in different scenarios in the field of medical image processing [8]. Among them, Convolutional Neural Networks (CNN) has shown superior performance in the image processing field, especially in extracting local features of images. More and more researchers [9–11] have developed new CNN models to discriminate the benign or malignant nature of pulmonary nodules in lung CT images. However, CNN performance is relatively weak to process global information. In particular, lung nodules in CT images are characterized by fuzzy boundaries and diverse shapes (as shown in Fig 1), which results in insufficient global feature extraction in many CNN networks during the classification of pulmonary nodules as benign or malignant, leading to poor accuracy. In contrast, the Transformer network [12] used in natural language processing excels at extracting long-distance contextual information and has been applied to image tasks in recent years. The Transformer network has a strong ability to process global information but is relatively weak to process local information. Since lung nodules have complex shapes and are prone to adhesion with lung tissues, in order to perform benign and malignant classification more accurately, this paper fuses CNN with Transformer-based ViT network [13], and proposes a ViT (Cross-ViT) network with cross-fusion attention. Cross-ViT network introduces the Cross- fusion Attention (CFA) module, which effectively captures and processes the local and global information of pulmonary nodules in CT images by fusing the features extracted by CNN and ViT, which improves the accuracy of classifying the benign and malignant nature of pulmonary nodules. The primary contributions of this paper are as follows:

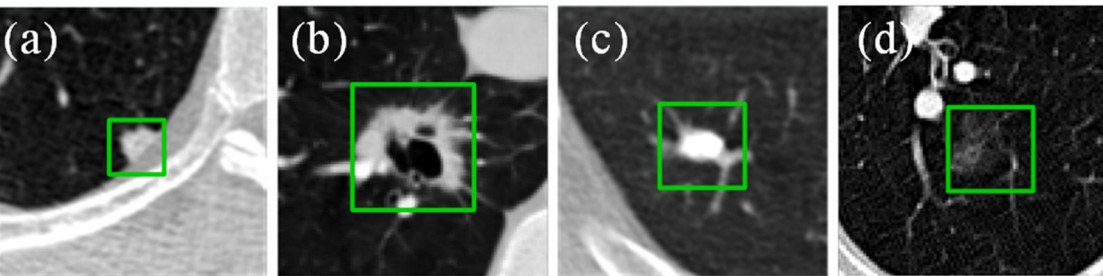

**Fig 1. Examples of lung nodules with various shapes in CT images.** (a) juxtapleural nodule. (b) cavitary nodule. (c) calcific nodule. (d) ground-glass opacity nodule.

1. Aiming at the problem of insufficient feature extraction in the existing classification models for benign and malignant pulmonary nodules, a Cross-ViT network that can fuse local and global information is proposed.

2. The feature Coupling (FC) module is proposed to solve the problem that CNN branch and Transformer branch feature map scales cannot be fused due to mismatch.

3. The CFA module is proposed to fuse local and global features in both directions.

The structure of the paper is structured as follows: The Material and methods chapter described the dataset and the structure of the network proposed in this paper, and the Result chapter described the details of the experiments and discussed the experimental results in detail. In the Conclusion chapter, the methods of this paper are summarized, and future research is prospected. Source code is available at https://github.com/tipyan/benign-and-malignant-classification.

## Material and methods

### Dataset

The experiments in this paper use CT images of lung nodules from the LUNA16 dataset to train and evaluate the proposed network. The LUNA16 dataset stands for Lung Nodule Analysis 16. This dataset was introduced in 2016 to develop a CAD system that can automatically detect lung nodules in CT scans. The LUNA16 dataset is a subset of the largest open-source pulmonary nodule dataset, LIDC-IDRI [14]. The LIDC-IDRI dataset is collected by the U.S. National Cancer Institute and has a total of 1,018 study instances. In the LUNA16 dataset, nodules less than 3 mm in diameter are called micronodules, and if the slices of a CT scan are too thick, there may be a situation in which the CT slice does not contain pixels of pulmonary nodules. Moreover, the malignancy rate of micronodules is extremely low, and it is not meaningful to categorize micronodules. Therefore, the LUNA16 dataset was screened against the LIDC-IDRI dataset, the final dataset contains 1,186 nodules.

The benign and malignant grades of pulmonary nodules were categorized into 5 categories, with category 1 being benign, category 2 being suspected benign, category 3 being uncertain, category 4 being suspected malignant, and category 5 being malignant. In this paper, we removed the uncertain nodules labeled as category 3 in the LUNA16 dataset, and then the nodules labeled as malignant or suspected malignant by all 4 physicians were recorded as malignant nodules with the label set to 1, and the remaining pulmonary nodule samples were

recorded as benign with the label set to 0. The imbalance of data would have an impact on the effectiveness of the network of classification. In unbalanced data, the results of model training tend to favor the side with the greater number. Therefore, the balance of data is very important for the task of pulmonary nodule classification. The dataset after screening and reformulation of labels contained a total of 1004 pulmonary nodules, of which 450 were malignant and 554 were benign, with a ratio of positive to negative samples close to 1:1.

Before the CT images are input into the network, appropriate cropping is beneficial to reduce the proportion of the image occupied by the non-pulmonary nodule background in the image, in order to enhance the attention of the classification network to the pulmonary nodules. The lung nodule classification task filtered out 1004 lung nodules, and the size statistics of these 1004 lung nodules are shown in Fig 2. From Fig 2, it can be seen that when the cropped size size is 35×35 basically covers all the nodule information. In order to include all the lung nodule information and facilitate network input, a CT data block of size $3 \times 48 \times 48$ is finally cropped with the nodule that is the center of the input data. Where 3 represents 3 layers of CT slices containing lung nodules, which are the center layer of the nodule slice and the upper and lower slices immediately above and below the center slice.

Since the cropped image is only 48×48×3 in size, a single pixel contains too much information, which is not conducive to the use of more complex preprocessing methods. Therefore,

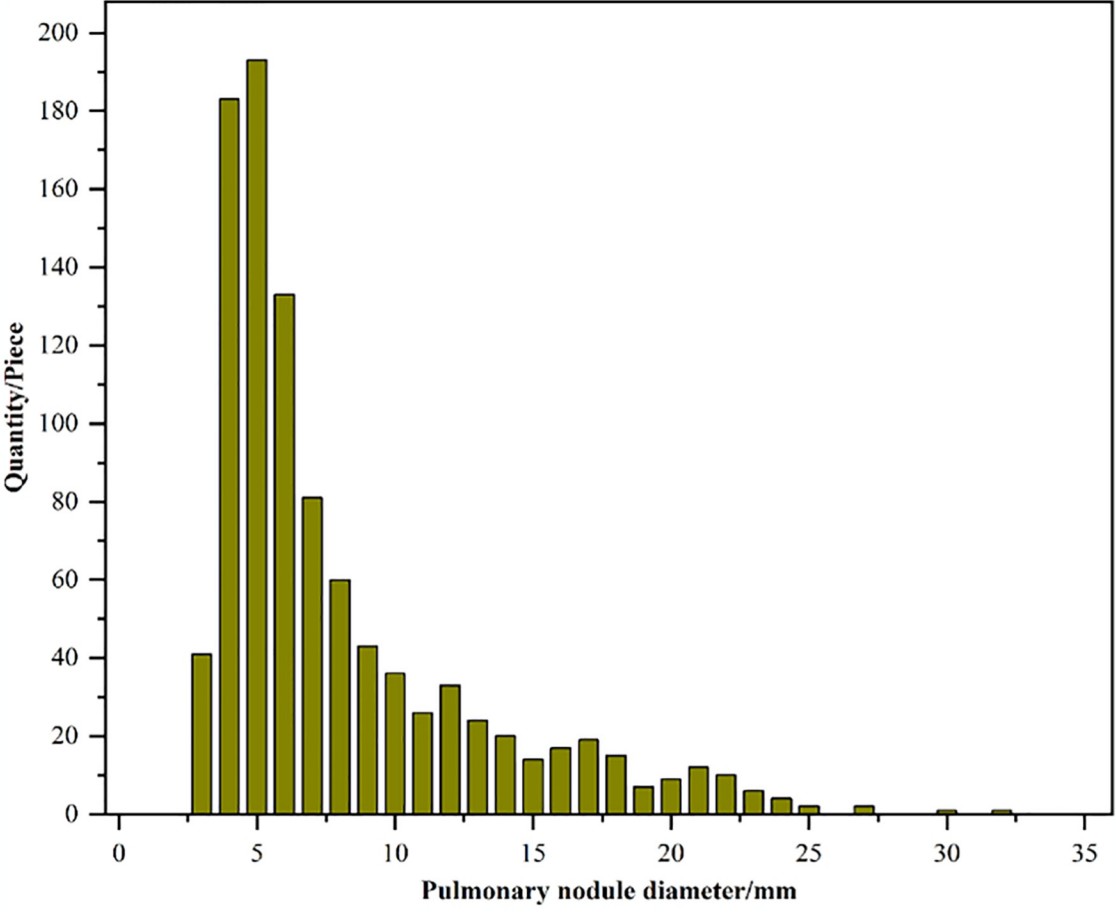

**Fig 2. Diameter distribution of pulmonary nodules after screening in the LUNA16 dataset.**

this paper only carried out normalization of the CT image simply. In addition, this paper also adds an online random flipping data enhancement method.

## Proposed method overview

Two key concepts are often involved when extracting visual features: local features and global features. Local features are small-area vector representations of an image, which plays a key role in many computer vision algorithms. Extracting local features helps to better understand the local information of an image. Global representations include some holistic information such as contours, shapes and object types of the image. Traditional CNN networks learn the local information of an image in a hierarchical manner through convolutional operations, while Transformer-based ViT networks integrate global information through cascading self-attentive modules.

Fig 3 shows examples of lung nodules of different sizes and shapes. When classifying these lung nodules, feature extraction of size and morphology should take into account the overall image, while the local image should be taken into account when extracting small features. Therefore, it is particularly important to integrate the local features of CNN and the global features of Transformer in the pulmonary nodule classification task.

In order to synthesize the advantages of the two networks, this paper proposes a Cross-ViT network with cross-confused attention for lung nodule classification, the structure is shown in Fig 4. The Cross-ViT network performs complementary fusion of information after extracting features from the CNN branch and the ViT branch, respectively, which fully combines the advantages of the two networks in extracting different features.

The Cross-ViT network mainly consists of two branches: a CNN branch for extracting local features and a Transformer branch for capturing global features. CT images of pulmonary nodules are input into each branch separately for feature extraction, and this parallel structure means that it maximizes the preservation of local and global features. However, the sizes of the feature maps output from the two are usually mismatched, so an FC module is needed to

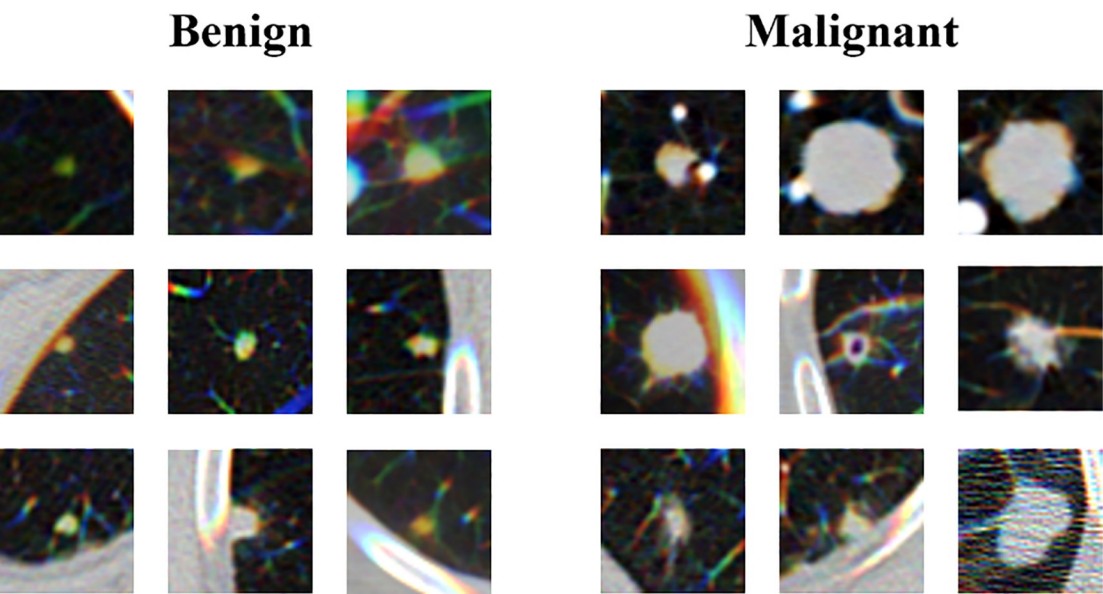

**Fig 3. Sections of lung nodules of different sizes and shapes.**

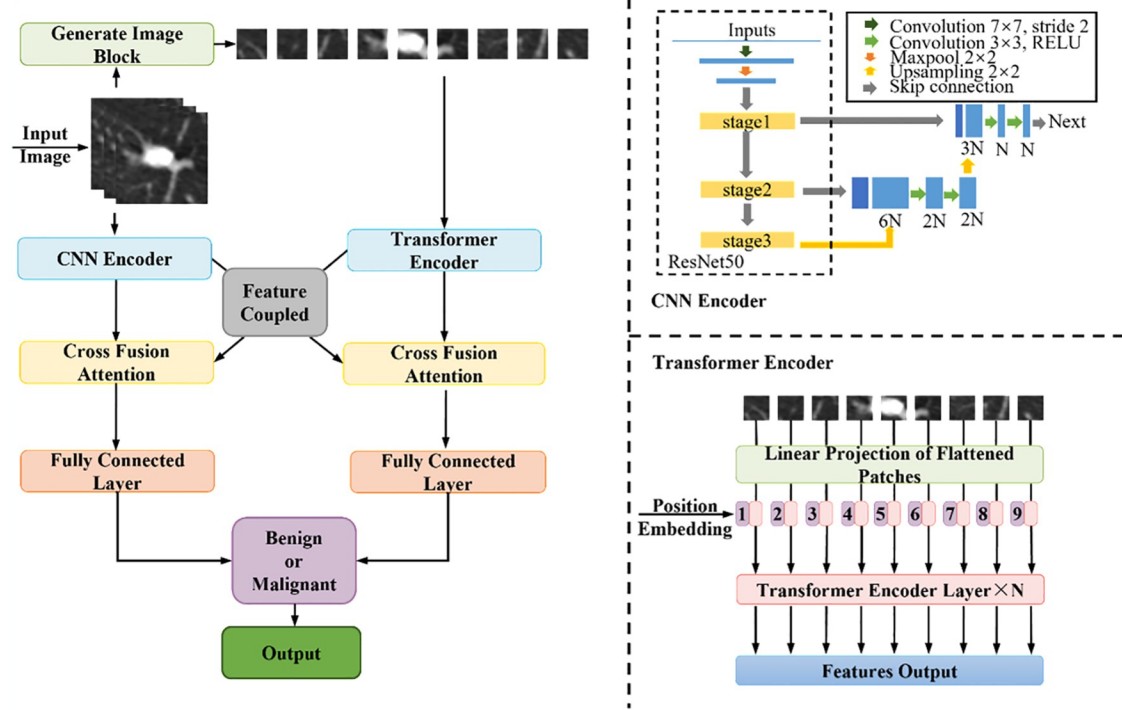

**Fig 4. Cross-fusion attention ViT network diagram.**

reshape the size between the output feature maps of the two branches for feature fusion by the CFA module. For the feature maps output from both branches, a fully connected layer serves as the classifier, and both branches are trained using a binary cross-entropy loss function. Finally, the mean of the probabilities of the two for classification is taken as the final classification result. The details of the structure of each component in the Cross-ViT network will be provided in the following subsections.

## CNN branch and Transformer branch of the proposed method

**CNN branch.** ResNet50 [15] introduces the residual structure, which can make the training of the network smoother and more stable. Therefore, the CNN branch uses ResNet50 to extract local features of pulmonary nodules. There are five stages in ResNet50. In Stage 0, the image is downsampled using a large 7×7 convolution to preserve the original image information as much as possible. The CNN branch inputs the image, and then downsamples with a 7×7 convolution to reduce the feature map size. Each of the latter four stages consists of an unequal bottleneck layer, each bottleneck layer consists of one 1×1 convolution, one 3×3 convolution, one 1×1 convolution, and the residual connection between the bottleneck input and output. To reduce the dimensionality of the ResNet50 output, only Stage1- Stage3 are taken as the feature extraction nets for the CNN branches.

As shown in Fig 4, we take the outputs of Stage1-Stage3 as the features extracted in the CNN part, and the outputs of Stage1-Stage3 contain information of different scales. On this basis, this paper adds the decoder part of Unet as the fusion of features of different scales. The output of stage 3 unsampled is combined with the output of stage 2, then combined with the output of stage 1 again after two convolution layers and upsampling, and output after two

convolution layers. Since the input first goes through a convolution with a kernel size of 7×7 and a stride of 2, and a maxpooling with a stride of 2, the input size of stage1 is 1/4 of the original size, and the output of the CNN part is the same as the output size of stage1, which is also 1/4 of the original size. This makes the output size of the CNN section the same as the output size of the subsequent Transformer section.

**Transformer branch.** The Transformer branch uses the ViT network to extract the global features of the pulmonary nodules. The ViT network is an image classification network proposed by Google team, and its structure largely retains the Transformer structure. The ViT structure is shown in Fig 5, it comprises five parts: Linear Projection of Flattened Patches, class token, Position Embedding, Transformer Encoder and MLP Head.

The Transformer input requirement is a sequence of vectors, while images are three channel pixel matrices, which does not meet the input requirement. Therefore, the images are cropped into a number of image blocks before input, and then these image blocks are linearly projected and straightened into one-dimensional vectors with classification markers and embedding position coding, which finally form Embedding Patches.

The ViT network uses several Transformer encoders to extract features. As can be seen in Fig 5, each Transformer encoder consists of layers Layer Normalization (LN), Multi-Head Attention, MLP block. LN is a normalization method introduced for Natural Language Processing, which avoids the occurrence of gradient vanishing phenomenon. After the data is normalized, it enters the Multi-Head Attention mechanism for computation. The multi-head attention mechanism can integrate the features learned by different heads and is the most important structure in the whole ViT network, whose structure is shown in Fig 6.

As can be seen in Fig 6, multiple heads in the multi-head attention mechanism enable the model to attend to various aspects of information. This mechanism stacks the Scaled Dot-Product Attention process h times and combining the outputs. After computing the scaled dot product attention process, three weights representing the query (Q), key (K), and value (V), respectively, are first extracted, where K has size $d_k$ and V has size $\sqrt{d_v}$. In this calculation process, Q is first subjected to a dot product operation with K, where each K is then divided by

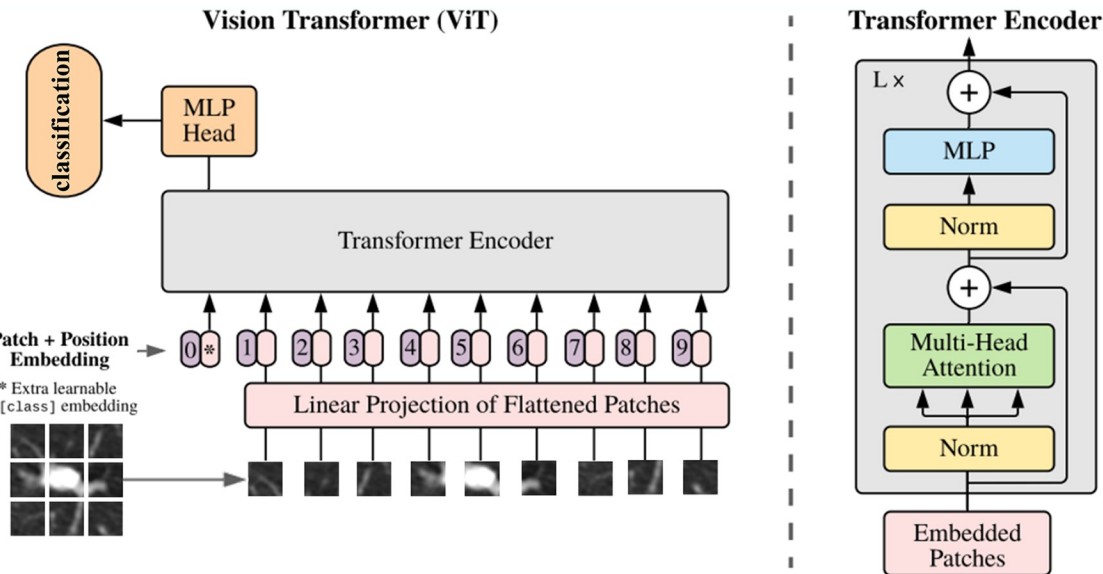

**Fig 5. Illustration of ViT network.**

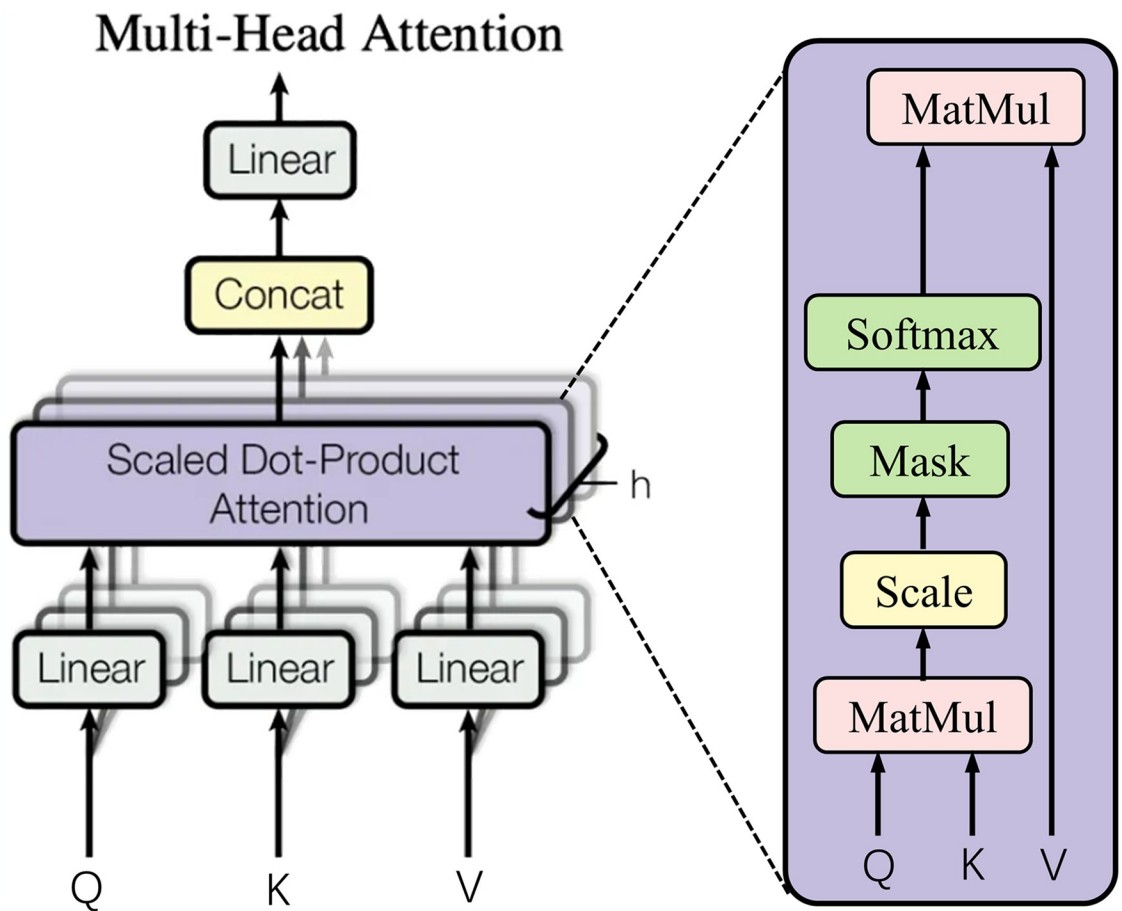

**Fig 6. Chart of multi-head attention mechanism structure.**

$\sqrt{d_v}$, followed by utilizing the Softmax function to obtain the weights of the V. The overall calculation of the single head attention is shown in Eq (1).

$$Attention(Q, K, V) = softmax\left(\frac{QK^T}{\sqrt{d_k}}\right)V \tag{1}$$

Single head attention does not pay enough attention to the model, resulting in extracting information poorly. In contrast, the basic idea of the multi-head attention mechanism is to notice the information from each representation subspace at the same time in each position, so that the manipulation of the different positional inputs is produced in an order independent contextual information attention, which is a global attention mechanism. The multi-head attention calculation process is shown in Eq (2).

$$MultiHead(Q, K, V) = Concat(head_1, head_2, \cdots, head_n)w^o \tag{2}$$

where *Concat* denotes matrix splicing, $head_i$ denotes the $i^{th}$ single-head attention, $W_i^Q \in \mathbb{R}^{d_{model} \times d_k}$, $W_i^K \in \mathbb{R}^{d_{model} \times d_k}$, $W_i^V \in \mathbb{R}^{d_{model} \times d_k}$, $W_i^O \in \mathbb{R}^{d_{model} \times d_k}$.

After passing through the multi-head attention mechanism, the input data proceeds to a MLP block where the input vector length is quadrupled in the fully connected layer, and the

vector is reshaped to the original length in the output, so that more deep information can be learned, and it is guaranteed that the size of the vector is the same. At this point, a Transformer encoder computation process is finished, such an operation will be repeated L times. Eventually the MLP header will integrate the information extracted earlier and then get the final classification result from the class token.

As shown in the Fig 4, the input size for the Transformer branch is $3 \times H \times W$ ($H$ and $W$ are the height and width of the input image, respectively), and it is first sliced in $3 \times 4 \times 4$ slices to get $H/4 \times W/4$ image blocks for subsequent operations. Same as ViT, this paper adopts the convolution method to slice images. The output channel size H of the convolution is 768, so the input size of Transformer branch is $H \times (H/4 \times W/4)$. The main purpose of Transformer branch in this article is to extract features, so the class token is removed from the method in this paper. After the position embedding, we use multiple consecutive Transformer encoder layers. In this experiment, the number of layers $N$ is 12, and each Transformer encoder layer contains 12 heads. The final output size of the Transformer branch is the same as the input size, which is $H \times (H/4 \times W/4)$.

## Feature Coupling module

The CNN branch and the Transformer branch output feature maps with different dimensions, and to realize the feature fusion of the two branches, the first step is to eliminate the difference in size between them. To address the issue of dimension mismatch between the two feature maps, this paper uses a two-way interactive FC module to match the size of the feature maps, and its specific structure is shown in Fig 7(a).

The output of CNN branch is a two-dimensional feature map, while the output of Transformer branch is a one-dimensional data, which has different dimensions. However, as described in the previous chapter, because CNN uses the decoder structure of Unet, the output of CNN branch is 1/4 of the original size (So we're going to write it in terms of $C \times H/4 \times W/4$.). For Tansformer branch, the output shape of the branch is the same as the input shape, while the input of Transformer branch is divided into the size of $E \times H/4 \times W/4$ from the original image and reshaped into the shape of $E \times N$ (where the size of N is HW/16). In order to integrate features extracted by two branchs, a Feature Coupling module is designed in this paper. As shown in Fig 7(a), the module processes the output of two branchs in different paths. The path from top to bottom is to process the output of CNN branch. The output of CNN branch goes through a $1 \times 1$ convolution and reshapes into the same size as the output of Transformer branch. This operation is to make the number of channels of the output of CNN branch consistent with Transformer branch. The bottom-to-top path is to process the output of the Transformer branch, which is similar to the processing of CNN branch. The output of Transformer branch is reshaped into the same shape as the output of CNN branch and integrated by $1 \times 1$ convolution. The results of both paths are used as the outputs of the FC module. The FC module achieves the two branches through a series of operations, which provides the basis for the subsequent feature fusion by the CFA module.

## Cross- fusion attention module

Non-local attention modules [16] are widely used as their ability to capture remote dependencies between locations. In order to fuse the feature maps output from the two branches, a cross- fusion attention module is proposed based on the non-local attention module, and its structure is shown in Fig 7(b).

CFA takes two inputs: one from the current branch's feature map and the other from a cross-branch feature map generated through the Feature Coupling module. When feature

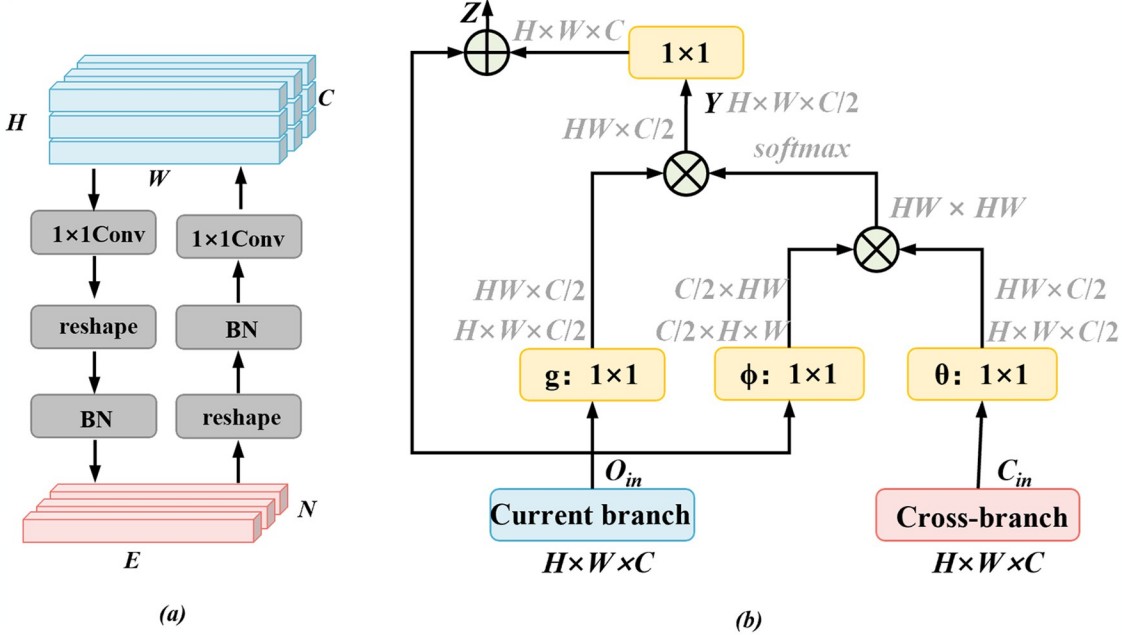

**Fig 7. The structure diagram of the two modules proposed in this paper.** (a) Feature Coupling module; (b) Cross-fusing attention module.

fusion is performed at the CNN branch, the size of the feature map $O_{in}$ at the CNN branch is H×W×C, and the size of the feature map $C_{in}$ at the integrated Transformer branch in the Feature Coupling module is H×W×C. The $O_{in}$ is first convolved with two *1 × 1* convolutions, the channel is reduced to C/2, and then a spreading operation is performed in both H×W dimensions to obtain a feature map of size H×W×C/2. The $C_{in}$ is convolved with one *1 × 1* convolution for the same purpose. Next the feature map passing through $\phi$ is timed with the feature map passing through $\theta$ to get a matrix of size HW×HW, and then the similarity matrix is calculated by softmax operation. Finally, the feature map passing through *g* is multiplied with the similarity matrix and the dimension is expanded to H×W×C/2 to get **Y**. The formula is shown in Eq (3).

$$y_i = softmax\left(\theta(x_i)^T\phi\left(x_j\right)\right)g\left(x_j\right) \tag{3}$$

where $x_i$ denotes the feature of $O_{in}$ at position $i$, $x_i$ denotes the feature of $C_{in}$ at position j, $g(x_j)$, $\phi(x_j)$ and $\theta(x_i)$ denotes the linear mapping (1×1 convolution can be used to instead), and $y_i$ denotes the output of the corresponding position **Y**.

The output **Y** passes through a 1×1 convolution to reshape the channels to C, and an elementwise summation operation is performed with the original output $O_{in}$ to obtain the output Z of the cross-fertilization attention module. When feature fusion is performed in the Transformer branch, the size of the feature map in the Transformer branch is E×N, and the size of the feature map in the integrated CNN branch in the Feature Coupling module input is E×N (E and N are the dimensionality of the feature vectors and the number of channels, respectively). In order to fuse the two features, the 1×1 convolution in the CFA module needs to be replaced with a 1D convolution before feature fusing.

The two branches perform feature fusion with each other and then output separately, such an operation can utilize the respective advantages of CNN and Transformer in feature extraction well and strengthen its classification capacity. The output of the network presented in this paper will contain the output of two branches.

$$loss_{diff} = BCE \begin{pmatrix} output_{CNN} \; if \; abs(output_{CNN} - 0.5) > \\ abs(output_{Trans} - 0.5) \; else \; output_{Trans} \end{pmatrix} \tag{4}$$

$$loss = loss_{diff} + BCE_{CNN} + BCE_{Trans} \tag{5}$$

As shown in the Eqs (4) and (5), *BCE* represents cross entropy loss, $output_{CNN}$ represents the output of CNN, and $output_{Trans}$ represents the output of Transformer. In the training stage, the output of each branch will calculate the difference with the label and retain the result with the furthest difference, then we calculate the crossentropy loss of the retained result and sum the crossentropy losses of the output result of each branch. The sum of the three will be the final loss of the method in this paper. In the test phase, the output of the two branches is farthest from 0.5 will be the final classification result.

## Results

### Experimental details

The experiments are conducted in an environment built on Windows 10 system, the processor used is 12700KF, the size of the running memory is 32GB, the graphics card is NVIDIA GeForce RTX 4080 with 16GB of video memory size, and the framework used is Pytorch 2.5. The dataset is randomly divided into two sets according to the ratio of 8:2, which are used for training and testing respectively. During the experiments, the batch size used for training is set to 32, optimization is performed using AdamW algorithm, the initial learning rate is set to 0.00002, the betas are set to (0.9, 0.999), and the total number of training rounds is 120.

The article also used the ExponentialLR method to update the learning rate in real time, which decreases the learning rate exponentially after each epoch training, and the parameter of gamma is set to 0.9 in this article. Fig 8 shows the change of the loss function with the epochs after using ExponentialLR or not. We can see that the loss updates are more stable when we are using ExponentialLR. The reduction of the volatility of the loss function also is helpful to avoid the occurrence of overfitting.

In order to evaluate the performance of the proposed network, a 5-fold cross-validation strategy is used for the experiments and the average segmentation performance is taken as the final segmentation result.

Operation time has always been the focus of deep learning-related work. The Cross-ViT proposed in this paper takes the image after slicing as input, and the size of the slice is only 48×48×3, which makes the calculation amount very small. In the experiment of this paper, the average calculation time of each graph is 6.34ms.

### Evaluation metrics

The task of classifying pulmonary nodules as benign or malignant is a dichotomous problem in which four cases occur in the specific classification result: true positive (TP), false positive (FP), true negative (TN), and false negative (FN). Classification metrics are capable of evaluating the performance of the model, and common classification metrics are Accuracy (ACC), Precision (PRE), Recall (REC), and F1-Score. These evaluation metrics can be calculated using TP, FP, TN, and FN.

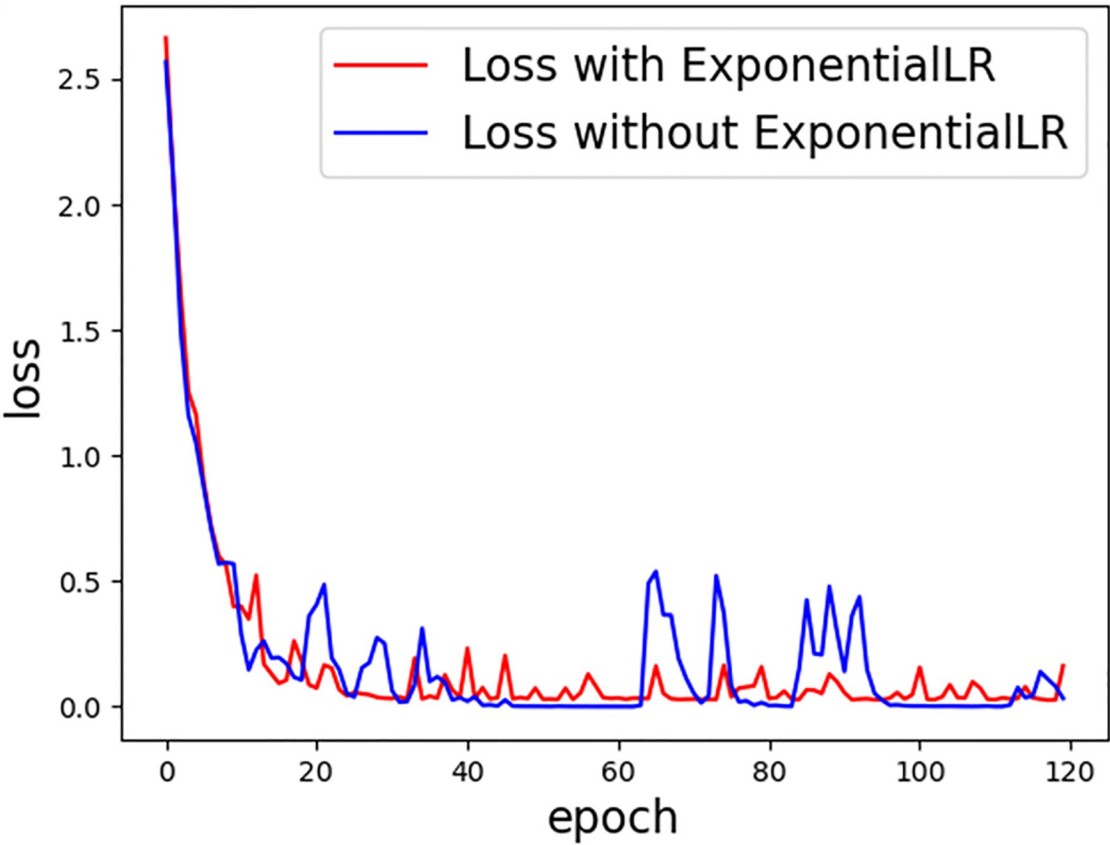

**Fig 8. Loss function changes with the epoch with ExponentialLR or without ExponentialLR.**

The formulas for calculating ACC, PRE, REC and F1-Score are given below:

$$ACC = \frac{TP + TN}{TP + TN + FP + FN} \tag{6}$$

$$PRE = \frac{TP}{TP + FP} \tag{7}$$

$$REC = \frac{TP}{TP + FN} \tag{8}$$

$$F1Score = \frac{2 \times PRE \times REC}{PRE + REC} \tag{9}$$

## 5-Fold cross-validation

In order to make effective use of the datasets, in this section, the Cross-ViT network is trained and validated using the 5-fold cross-validation method, the experimental results are presented in Table 1. It can be seen that the ACC, PRE, REC and F1-score of the results of the 5-fold cross-validation reach 91.04%, 91.96%, 92.45% and 91.92%, respectively.

**Table 1. Results of cross-validation of the Crdoss ViT network for benign and malignant pulmonary nodules.**

| Folds | ACC(%) | PRE(%) | REC(%) | F1-Score (%) |
|---|---|---|---|---|
| Fold1 | 91.05 | 92.86 | 91.23 | 92.04 |
| Fold2 | 89.05 | 91.07 | 89.47 | 90.26 |
| Fold3 | 92.51 | 91.89 | 94.44 | 93.15 |
| Fold4 | 89.55 | 89.30 | 91.74 | 90.51 |
| Fold5 | 93.03 | 91.96 | 95.37 | 93.63 |
| Average | 91.04 | 91.42 | 92.45 | 91.92 |

## Ablation experimental validation

To verify the validity of the FC and CFA modules proposed in this paper, we tested the ablation experiments without the addition of these two modules. In the ablation experiment in this paper, output features of CNN branch and Transformer branch are directly output without fusion, and other settings are the same as in this paper.

Table 2 lists the results of using FC and CFA modules and without using FC and CFA modules. It can be seen that except ACC, the PRE, REC and F1-Score using FC and CFA modules are 0.96%, 3.12% and 2.03% higher respectively, while ACC index is only 0.31% lower than that without FC and CFA modules. Kappa consistency test was also carried out in this paper, and it can be seen that both with and without FC and CFA have high consistency, but the consistency of the method with FC and CFA is low but similar. Combining PRE and REC, it can be seen that the proposed method has higher recognition accuracy for positive samples and can have a good fusion effect on the features extracted by the two branches.

## Comparative experimental validation

To further validate the performance of the proposed Cross-ViT network, this section compares the performance of the proposed method with some current mainstream classification networks and lung nodule classification networks, and the results are displayed in Table 3 and Fig 9.

It can be seen that VGG16 [17] has the worst performance in all aspects, and its redundant parameters can easily lead to the overfitting problem when training on small data. GoogleNet [18] and ResNet50 have improved accuracy compared to VGG16, and their accuracies reach 88.36% and 87.54%, respectively. Compared with VGG16, GoogleNet and ResNet50 have fewer parameters and deeper networks, which can achieve better results with fewer training parameters, but still fail to capture important information when extracting complex CT images. SENet [19] adds the SE attention module compared with ResNet50, which pays more attention to the lung nodule, and the accuracy was improved to 88.24%. ViT networks do not have a feature modeling structure similar to that of CNN networks that deepens layer by layer to capture the local features of lung nodules, and thus their classification ability does not have a great advantage over that of CNNs. These networks do not set a more favorable feature

**Table 2. Results of ablation experiments without FC and CFA modules.**

| Methods | ACC(%) | PRE(%) | REC(%) | F1-Score (%) | Kappa |
|---|---|---|---|---|---|
| Without FC and CFA | 91.35 | 90.46 | 89.33 | 89.89 | 0.8252 |
| With FC and CFA | 91.04 | 91.42 | 92.45 | 91.92 | 0.8186 |

**Table 3. Comparison of pulmonary nodule classification networks.**

| Methods | ACC(%) | PRE(%) | REC(%) | F1-Score (%) |
|---|---|---|---|---|
| VGG16 [17] | 86.43 | 89.41 | 87.23 | 88.31 |
| GoogleNet [18] | 87.78 | 90.89 | 89.65 | 90.27 |
| ResNet50 | 87.54 | 90.23 | 88.47 | 89.34 |
| SENet [19] | 88.24 | 90.77 | 88.56 | 89.65 |
| ViT | 88.23 | 90.35 | 90.27 | 90.31 |
| DeepLung [20] | 90.44 | - | **95.80** | - |
| 3D-SE-CDNet [21] | 89.93 | - | 83.34 | - |
| DFOF [22] | 92.13 | 94.16 | 87.16 | 89.93 |
| Tang et al. [23] | 89.35 | - | 87.31 | - |
| Lin et al. [24] | **92.80** | - | - | 92.16 |
| Cross-ViT | 91.04 | 91.42 | 92.45 | 91.92 |
| Cross-ViT (SENet) | 92.43 | **94.27** | 91.68 | **92.96** |

extraction structure for lung nodule images, so this section also compares them with some network models in the field of pulmonary nodule classification.

Notably, the Cross-ViT network proposed in this paper outperforms most of the methods in terms of performance. This is because the Cross-ViT network fully combines the ability of the CNN network for extracting local features with the ability of the ViT network for extracting context-dependent information, which results in a better performance, but some of the performance metrics are not as superior as those of DFOF. Considering that the CNN branch uses ResNet50, which may be insufficient for extracting information for complex images such as lung nodules, this paper re-tested the performance after changing the CNN branch to

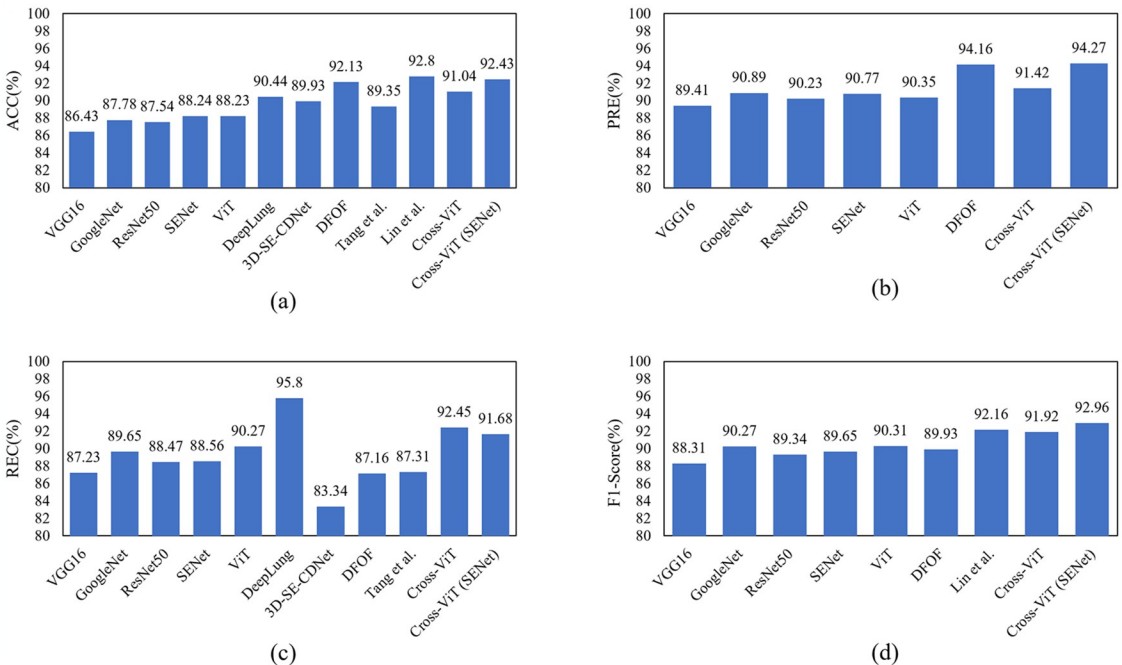

**Fig 9. Histograms of each metric compared to state of the art methods.** (a) ACC; (b) PRE; (c) REC; (d) F1-Score.

SENet and found that the performance of the improved model is even more superior, exceeding that of the DFOF model. Compared with the results of Lin et al., although the ACC index of the proposed method is lower, the F1-Score of this paper is higher. In general, the results of this method are similar to those of Lin et al.

## Limitations

Although the method in this paper integrates CNN branch and Transformer branch at the feature layer, the output of each branch is still retained in the output results. For these two outputs, the most likely part is selected in the method of this paper (results greater than 0.5 are calculated as the probability of a positive sample, and samples less than 0.5 are calculated as the probability of a negative sample). In the future work, a more intelligent result selection method can be studied to achieve more accurate threshold division and result output.

## Conclusion

In this paper, to tackle the issue of insufficient feature extraction in the classification model for benign and malignant lung nodules, a Cross-ViT network is proposed to fuse the benign and malignant nature of pulmonary nodules. Cross-ViT network uses CNN branch and Transformer branch to extract the local and global features of the lung nodules respectively, and it fuses the two kinds of features to discriminate the benign and malignant nature to a certain extent. The results show that the Cross-ViT network proposed in this paper is superior for classifying the lung nodules as benign or malignant. In future work, we intend to assess the network on additional medical image datasets and refine the fusion of local and global features according to classification outcomes across different datasets to improve generality and accuracy.

## Author Contributions

**Conceptualization:** Qinfang Zhu, Liangyan Fei.

**Formal analysis:** Qinfang Zhu, Liangyan Fei.

**Methodology:** Liangyan Fei.

**Writing – original draft:** Liangyan Fei.

**Writing – review & editing:** Qinfang Zhu.

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
