## [Decision Letter · Decision Letter 0]

4 Oct 2024

PONE-D-24-34082Cross-ViT based benign and malignant classification of pulmonary nodulesPLOS ONE

Dear Dr. Zhu,

Thank you for submitting your manuscript to PLOS ONE. After careful consideration, we feel that it has merit but does not fully meet PLOS ONE’s publication criteria as it currently stands. Therefore, we invite you to submit a revised version of the manuscript that addresses the points raised during the review process.

We look forward to receiving your revised manuscript.

Kind regards,

Sarada Prasad Dakua

Academic Editor

PLOS ONE

Journal Requirements:

Additional Editor Comments:

The authors are advised to carefully address the reviewers' comments.

Reviewers' comments:

Reviewer's Responses to Questions

**Comments to the Author**

1. Is the manuscript technically sound, and do the data support the conclusions?

Reviewer #1: Yes

Reviewer #2: Yes

2. Has the statistical analysis been performed appropriately and rigorously? 

Reviewer #1: N/A

Reviewer #2: No

3. Have the authors made all data underlying the findings in their manuscript fully available?

Reviewer #1: No

Reviewer #2: No

4. Is the manuscript presented in an intelligible fashion and written in standard English?

Reviewer #1: Yes

Reviewer #2: Yes

5. Review Comments to the Author

Reviewer #1: The paper has addressed a nice research problem; however, I have the below comments: when applying the proposed Cross-ViT network for benign and malignant classification of pulmonary nodules, there seems several limitations that the authors need to discuss in their revision:

1. The performance of Cross-ViT is highly reliant on the quality and size of the training dataset. If the dataset lacks sufficient examples of either benign or malignant nodules, the model may not generalize well.

2. Medical datasets often suffer from class imbalance, which can lead to biased predictions favoring the more prevalent class. This is critical in nodule classification, where benign cases might outnumber malignant ones.I would suugest the authors to discuss by citing the below papers

“Perspectives on the technological aspects and biomedical applications of Virus-like-particles/ Nanoparticles in reproductive biology: Insights on the medicinal and toxicological outlook,” Advanced NanoBiomed Research, Wiley, 2:2200010, 2022.

"Synergistic and Additive Effects of Menadione in Combination with Antibiotics on Multidrug-Resistant Staphylococcus aureus: Insights from Structure-Function Analysis of Naphthoquinones," ChemMedChem, Chemistry Eurpoe vol. 18, no. 24, 2023.

"Leveraging hallmark Alzheimer’s molecular targets using phytoconstituents: Current perspective and emerging trends,"Biomedicine & Pharmacotherapy, Elsevier, vol. 139, no. 111634, 2021

3. The integration of local features from CNNs with global features from Transformers may not always align well in the context of medical images, potentially leading to suboptimal feature fusion and classification performance.

4. The complex architecture can lead to overfitting, especially when trained on limited datasets, which is common in medical imaging.

5. The model's architecture requires significant computational resources, which may not be feasible in clinical settings with limited hardware. Complexity remains an issue, the authors can refer the below papers while discussing this: “Real-time Automated Image Segmentation Technique for Cerebral Aneurysm on Reconfigurable System-On-Chip,” Journal of Computational Science, Elsevier, vol. 27, pp 35-45, 2018.

“Lattice-Boltzmann Interactive Blood Flow Simulation Pipeline,” International Journal of Computer Assisted Radiology and Surgery, Springer, vol.15, pp. 629-639, 2020.

“Zynq SoC based Acceleration of the Lattice Boltzmann Method,” Concurrency and Computation: Practice and Experience, Wiley, col. 31, issue 17, 2019.

“Heterogeneous System-on-Chip based Lattice- Boltzmann Visual Simulation System,” Systems Journal, IEEE, vol. 14, no. 2, pp. 1592-1601, 2020

6. Medical images can contain artifacts or noise, and the attention mechanisms may not always effectively filter out irrelevant information, impacting classification accuracy.

7. Effective feature extraction relies on the quality of preprocessing steps (e.g., normalization, augmentation). Poor preprocessing can adversely affect both local and global feature extraction. The authors could cite the below papers and discuss if pre-processing could be of help: “A Lightweight Neural Network with Multiscale Feature Enhancement for Liver CT Segmentation,” Scientific Reports, Nature, vol. 12, no. 14153, pp. 1-12, 2022.

“Re- routing drugs to blood brain barrier: A comprehensive analysis of Machine Learning approaches with fingerprint amalgamation and data balancing,” IEEE Access, vol. 11, pp. 9890-9906, 2023.

“Dense-PSP-UNet: A Neural Network for Fast Inference Liver Ultrasound Segmentation,” Computers in Biology and Medicine, ScienceDirect, vol. 153, pp. 106478, 2023.

8. Please include the potential limitations of the paper.

Reviewer #2: 1. Many methods have used CNNs for preliminary feature extraction and then Vision Transformers (ViTs) for capturing long-range dependencies in the feature maps. How is this work different from the other works that have been proposed for medical image segmentation?

2. Authors have mentioned performance on the LUNA 16 dataset at the end of the abstract, but how much is the improvement over the best-performing state-of-the-art model compared to the work proposed in this paper? This should be highlighted numerically at the end of the abstract.

3. Lines 50 to 54 claim CNNs have been used. More references specifically for applications should be cited here. Some of them that should be included are as follows:

a. A lightweight neural network with multiscale feature enhancement for liver CT segmentation

b. A lightweight neural network with multiscale feature enhancement for liver CT segmentation

c. Enhancing ECG-based heart age: impact of acquisition parameters and generalization strategies for varying signal morphologies and corruptions

d. Advancements in Deep Learning for B-Mode Ultrasound Segmentation: A Comprehensive Review

e. Unveiling the future of breast cancer assessment: a critical review on generative adversarial networks in elastography ultrasound

f. Neural network-based fast liver ultrasound image segmentation

4. Add the structure of the paper at the end of the introduction.

5. Explanation of local and global features in lines 137 to 142 should be improved by explaining them with specific examples in the medical images that the work is based on.

6. Lines 142 to 150 reiterate what has already been stated in the introduction regarding CNN transformers. This section of the network overview should cover the overall architecture of the CrossVIT model, not reintroduce the contributions of the paper.

7. The size difference between the CNN encoder and the transformer encoder is not the only problem. These two encoders are capturing features at different scales. How is the feature coupling block handling the feature maps that are generated at completely different scales?

8. How can the probability generated from the two branches be summed? This can ruin the probability distribution, as the result may not remain between 0 and 1. This approach needs to be clarified or corrected.

9. How many transformer blocks were used? What is the configuration of the transformer block in terms of the number of heads and other hyperparameters? More details regarding the hyperparameters and the configuration of both the CNN and transformer branches need to be provided.

10. The textual description of the feature coupling module does not correspond to the diagram shown. Authors talk about feedback between the branches. What do they mean by this? Please rewrite the entire description of the feature coupling module for better clarity.

11. According to the CFA diagram, the cross-branch feature map serves as the query, and the current branch provides the key and the value. Is that correct? If so, please clarify why this design choice was made for the cross-fusion attention module.

12. Add a GitHub link with proper documentation explaining the network. It should contain all the necessary files to reproduce the results on the LUNA 16 dataset for this network.

13. The paper lacks ablation studies to show the performance impact of cross-vision attention and feature coupling, and the role of individual encoders. Ablation studies are necessary to show the performance contribution of each component.

14. The network has been compared with fairly old architectures like VGG-16, GoogleNet, ResNet, and ViTs. Please compare it with more state-of-the-art architectures from the literature for a fairer comparison.

15. There is no discussion on the limitations of this work. Please include a limitations section in the paper, discussing any potential shortcomings or areas where the method could be improved.

6. PLOS authors have the option to publish the peer review history of their article (what does this mean?). If published, this will include your full peer review and any attached files.

Reviewer #1: No

Reviewer #2: No

---

## [Author Response · Author response to Decision Letter 0]

8 Nov 2024

Response to the comments

Dear Editor and Reviewer:

Thank you for your letter and the reviewer’s comments concerning our manuscript entitled Cross-ViT based benign and malignant classification of pulmonary nodules. Those comments are all valuable and very helpful for revising and improving our paper, we have studied comments carefully and made corrections which we hope to meet with approval. And we carefully listened to the reviewers and added a new table (Table 2), four new pictures (Fig 2,4,7,8,11) and replaced some tables and pictures (Fig 8, Table 3). In addition, we have carefully checked and modified some of the content (marked red in text) in the paper to make the article more rigorous. The responses to the reviewers’ comments are as follows.

Reviewer #1: The paper has addressed a nice research problem; however, I have the below comments: when applying the proposed Cross-ViT network for benign and malignant classification of pulmonary nodules, there seems several limitations that the authors need to discuss in their revision:

1. The performance of Cross-ViT is highly reliant on the quality and size of the training dataset. If the dataset lacks sufficient examples of either benign or malignant nodules, the model may not generalize well.

[Response]: Thank you so much for your suggestion. The method proposed in this paper is based on deep learning, and the development of deep learning inevitably needs to rely on big data. Many of today's mature deep learning models are supported by large amounts of data, such as chatgpt. How to train better results with less data and smaller models is also the focus of current research in the field of deep learning. In the method presented in this paper, we validate our results using a publicly available data set, and for further generalization problems, a larger data set should be reconstructed in real applications to retrain the model.

2. Medical datasets often suffer from class imbalance, which can lead to biased predictions favoring the more prevalent class. This is critical in nodule classification, where benign cases might outnumber malignant ones.I would suugest the authors to discuss by citing the below papers

“Perspectives on the technological aspects and biomedical applications of Virus-like-particles/ Nanoparticles in reproductive biology: Insights on the medicinal and toxicological outlook,” Advanced NanoBiomed Research, Wiley, 2:2200010, 2022.

"Synergistic and Additive Effects of Menadione in Combination with Antibiotics on Multidrug-Resistant Staphylococcus aureus: Insights from Structure-Function Analysis of Naphthoquinones," ChemMedChem, Chemistry Eurpoe vol. 18, no. 24, 2023.

"Leveraging hallmark Alzheimer’s molecular targets using phytoconstituents: Current perspective and emerging trends,"Biomedicine & Pharmacotherapy, Elsevier, vol. 139, no. 111634, 2021

[Response]: Thank you so much for your suggestion. Our data contained a total of 1004 data, of which 450 were malignant and 554 were benign. The data are almost in balance. We added relevant instructions and references to the literature in the Dataset section.

3. The integration of local features from CNNs with global features from Transformers may not always align well in the context of medical images, potentially leading to suboptimal feature fusion and classification performance.

[Response]: Thank you so much for your suggestion. The CNN network used in this paper takes resnet as the backbone and takes the output of stage1-3 as the extracted features. Then, the decoder part of unet is also quoted in this paper to fuse the features of different scales in stage 1-3. Since the input of resnet will first go through a 7×7 convolution with stride 2 and a maxpool with stride 2, the output size of CNN in this paper is 1/4 of the original image, while the input and output of Transformer are also 1/4 of the original image. Therefore, input scales of CNN branch and Transformer branch in the Feature Coupling module are the same. We have made some changes to the description of the method section.

4. The complex architecture can lead to overfitting, especially when trained on limited datasets, which is common in medical imaging.

[Response]: Thank you so much for your suggestion. Because the information contained in the limited data is not comprehensive, the overfitting problem is inevitable in the training of neural networks. To avoid overfitting, we applied ExponentialLR in our training, which uses exponential-adjusted learning rates that decline exponentially with each epoch. We added the training loss change graph to the Implementation details section. We can see that ExponentialLR effectively reduces the fluctuation of training losses.

5. The model's architecture requires significant computational resources, which may not be feasible in clinical settings with limited hardware. Complexity remains an issue, the authors can refer the below papers while discussing this: “Real-time Automated Image Segmentation Technique for Cerebral Aneurysm on Reconfigurable System-On-Chip,” Journal of Computational Science, Elsevier, vol. 27, pp 35-45, 2018.

“Lattice-Boltzmann Interactive Blood Flow Simulation Pipeline,” International Journal of Computer Assisted Radiology and Surgery, Springer, vol.15, pp. 629-639, 2020.

“Zynq SoC based Acceleration of the Lattice Boltzmann Method,” Concurrency and Computation: Practice and Experience, Wiley, col. 31, issue 17, 2019.

“Heterogeneous System-on-Chip based Lattice- Boltzmann Visual Simulation System,” Systems Journal, IEEE, vol. 14, no. 2, pp. 1592-1601, 2020

[Response]: Thank you so much for your suggestion. The main task of this paper is the classification of pulmonary nodules. The input of the method in this paper is the sliced pulmonary nodules image, the size of which is only 48×48×3, which is quite small compared with other objects, resulting in a very small computational load for the network with this image as input. In the experiment in this paper, the average time for processing a graph is 6.34ms. We have added references and explanations in the Implementation details section.

6. Medical images can contain artifacts or noise, and the attention mechanisms may not always effectively filter out irrelevant information, impacting classification accuracy.

[Response]: Thank you so much for your suggestion. Artifacts and noise can also be learned from the training of the network. The method in this paper integrates CNN and Transformer, so that local features and global features can be considered in our method at the same time. Similarly, artifacts and noise are classified as global and local, and the methods in this paper can also consider global and local noise more fully. We have revised the method section of this paper to make it more accurate.

7. Effective feature extraction relies on the quality of preprocessing steps (e.g., normalization, augmentation). Poor preprocessing can adversely affect both local and global feature extraction. The authors could cite the below papers and discuss if pre-processing could be of help: “A Lightweight Neural Network with Multiscale Feature Enhancement for Liver CT Segmentation,” Scientific Reports, Nature, vol. 12, no. 14153, pp. 1-12, 2022.

“Re- routing drugs to blood brain barrier: A comprehensive analysis of Machine Learning approaches with fingerprint amalgamation and data balancing,” IEEE Access, vol. 11, pp. 9890-9906, 2023.

“Dense-PSP-UNet: A Neural Network for Fast Inference Liver Ultrasound Segmentation,” Computers in Biology and Medicine, ScienceDirect, vol. 153, pp. 106478, 2023.

[Response]: Thank you so much for your suggestion. The image used in this paper is a slice of the lung nodule, whose size is only 48×48×3. Such a narrow resolution makes each pixel contain more information than a large image. However, overly complex preprocessing will destroy the structure between pixels, so it is not suitable to use more complex preprocessing methods in the case of large information density. Therefore, in the method proposed in this paper, we simply use the normalization of CT images and the online random flipping data enhancement method. We added the relevant instructions in the Dataset section.

8. Please include the potential limitations of the paper.

[Response]: Thank you so much for your suggestion. We have added limitations in the Experiments and discussion section.

Reviewer #2:

1. Many methods have used CNNs for preliminary feature extraction and then Vision Transformers (ViTs) for capturing long-range dependencies on the feature maps. How is this work different from the other works that have been proposed for medical image segmentation?

[Response]: Thank you so much for your suggestion. This paper proposes a Cross-ViT network that integrates CNN and transformer. The fusion network in this paper fuses features extracted from CNN and transformer, which has not been studied in previous pulmonary nodule classification works. At the beginning of the network, inputs are fed into the CNN branch and transformer branch respectively. After extracting the features of each branch, this paper designs a unique Feature Coupling module and a Cross-fusing attention to fuse the features of different branches.

2. Authors have mentioned performance on the LUNA 16 dataset at the end of the abstract, but how much is the improvement over the best-performing state-of-the-art model compared to the work proposed in this paper? This should be highlighted numerically at the end of the abstract.

[Response]: Thank you so much for your suggestion. We made changes to the summary as required.

3. Lines 50 to 54 claim CNNs have been used. More references specifically for applications should be cited here. Some of them that should be included are as follows:

a. A lightweight neural network with multiscale feature enhancement for liver CT segmentation

b. A lightweight neural network with multiscale feature enhancement for liver CT segmentation

c. Enhancing ECG-based heart age: impact of acquisition parameters and generalization strategies for varying signal morphologies and corruptions

d. Advancements in Deep Learning for B-Mode Ultrasound Segmentation: A Comprehensive Review

e. Unveiling the future of breast cancer assessment: a critical review on generative adversarial networks in elastography ultrasound

f. Neural network-based fast liver ultrasound image segmentation

[Response]: Thank you so much for your suggestion. We add more literature reviews on the application of CNN in the medical field in lines 54 to 62.

4. Add the structure of the paper at the end of the introduction.

[Response]: Thank you so much for your suggestion. We have added the structure of this article at the end of the Introduction chapter.

5. Explanation of local and global features in lines 137 to 142 should be improved by explaining them with specific examples in the medical images that the work is based on.

[Response]: Thank you so much for your suggestion. We have added a new example figure of lung nodules to the Network overview section, along with an explanation of the global and local features of lung nodules.

7. The size difference between the CNN encoder and the transformer encoder is not the only problem. These two encoders are capturing features at different scales. How is the feature coupling block handling the feature maps that are generated at completely different scales?

[Response]: Thank you so much for your suggestion. The output of STAGE1-3 of ResNet in CNN is used as the features in this paper, and the output of stage3 will be upsampled and combined with the output of stage2, and then the combined result will be convolved and upsampled and combined with the output of stage1, and finally the result will be output through convolution. This structure is similar to Unet, but since the input of ResNet will undergo a 7×7 convolution with stride 2 and a maximum pooling layer, the output of the input after passing through the CNN part will be 1/4 of the original size, and the input of Transformer part will also be 1/4. In other words, the output scale of CNN and Transformer is the same. We have added new instructions in the CNN branch section.

8. How can the probability generated from the two branches be summed? This can ruin the probability distribution, as the result may not remain between 0 and 1. This approach needs to be clarified or corrected.

[Response]: Thank you so much for your suggestion. The output of the network presented in this article will contain the output of two branches. In the training stage, the output of each branch will calculate the difference with the label, retain the result with the furthest difference, calculate the crossentropy loss of the retained result, and sum the crossentropy losses of the output result of each branch. The sum of the three will be the final loss of the method in this paper. In the test phase, the output of the two branches is farthest from 0.5 will be the final classification result. We have added new instructions and formulas at the end of the Cross-fusion Attention module chapter.

9. How many transformer blocks were used? What is the configuration of the transformer block in terms of the number of heads and other hyperparameters? More details regarding the hyperparameters and the configuration of both the CNN and transformer branches need to be provided.

[Response]: Thank you so much for your suggestion. This paper uses 12 consecutive transformer encoder layers, each containing 12 headers. We have made changes to the explanation of the Transformer branch section.

10. The textual description of the feature coupling module does not correspond to the diagram shown. Authors talk about feedback between the branches. What do they mean by this? Please rewrite the entire description of the feature coupling module for better clarity.

[Response]: Thank you so much for your suggestion. The feature coupling module is designed to fuse the output of the CNN branch to the Tranformer branch and the output of the Tranformer branch to the CNN branch. Therefore, the output of different branches in this module is processed similarly, so that the output of different branches can be merged into the relative branch in later operations. We have rewritten parts of the Feature Coupling module chapter.

11. According to the CFA diagram, the cross-branch feature map serves as the query, and the current branch provides the key and the value. Is that correct? If so, please clarify why this design choice was made for the cross-fusion attention module.

[Response]: Thank you so much for your suggestion. The cross-fusing attention module proposed in this paper is applied to CNN branch and Transformer branch respectively. The main function of this module is to fuse relative branch information in each branch. The relative branch information is obtained from the output of the Feature Coupling module, as shown in Figure 6. In the cross-fusing attention module of each branch, the current branch acts as the main body, while the output of the Feature Coupling module acts as supplementary information. Therefore, the current branch acts as k and v in this paper. The output of the Feature Coupling module is q.

12. Add a GitHub link with proper documentation explaining the network. It should contain all the necessary files to reproduce the results on the LUNA 16 dataset for this network.

[Response]: Thank you so much for your suggestion. We provide a link to the code at the end of the Introduction.

13. The paper lacks ablation studies to show the performance impact of cross-vision attention and feature coupling, and the role of individual encoders. Ablation studies are necessary to show the performance contribution of each component.

[Response]: Thank you so much for your suggestion. The results of ablation Experiments without FC and CFA modules were added to the Experiments and discussion section. Since FC and CFA modules are bound to each other, only one set of ablation experiments was conducted.

14. The network has been compared with fairly old architectures like VGG-16, GoogleNet, ResNet, and ViTs. Please compare it with more state-of-the-art architectures from the literature for a fairer comp

---

## [Editor Report · Decision Letter 1]

6 Dec 2024

PONE-D-24-34082R1Cross-ViT based benign and malignant classification of pulmonary nodulesPLOS ONE

Dear Dr. Zhu,

Thank you for submitting your manuscript to PLOS ONE. After careful consideration, we feel that it has merit but does not fully meet PLOS ONE’s publication criteria as it currently stands. Therefore, we invite you to submit a revised version of the manuscript that addresses the points raised during the review process.

We look forward to receiving your revised manuscript.

Kind regards,

Mohammad Amin Fraiwan

Academic Editor

PLOS ONE

Additional Editor Comments:

**As noted previously by the Journal Office, in the previous round of review the reviewers recommended that you cite specific previously published works. Members of the editorial team have determined that the works referenced are not directly related to the submitted manuscript. As such, please note that it is not necessary or expected to cite the works requested by the reviewers, and we suggest that you remove references 6-9 and 22-30, unless you feel they are particularly relevant to your manuscript. **

---

## [Author Response · Author response to Decision Letter 1]

8 Dec 2024

Response: Thank you for your letter and the comment concerning our manuscript entitled Cross-ViT based benign and malignant classification of pulmonary nodules. Your comment is valuable and very helpful for revising and improving our paper, we have carefully considered your concerns about references and have removed some of the references added in the last revision. Specifically, we deleted the references from the original labels [6-10] and [22-30].

---

## [Decision Letter · Decision Letter 2]

12 Jan 2025

PONE-D-24-34082R2Cross-ViT based benign and malignant classification of pulmonary nodulesPLOS ONE

Dear Dr. Zhu,

Thank you for submitting your manuscript to PLOS ONE. After careful consideration, we feel that it has merit but does not fully meet PLOS ONE’s publication criteria as it currently stands. Therefore, we invite you to submit a revised version of the manuscript that addresses the points raised during the review process.

We look forward to receiving your revised manuscript.

Kind regards,

Mohammad Amin Fraiwan

Academic Editor

PLOS ONE

Journal Requirements:

Reviewers' comments:

Reviewer's Responses to Questions

**Comments to the Author**

1. If the authors have adequately addressed your comments raised in a previous round of review and you feel that this manuscript is now acceptable for publication, you may indicate that here to bypass the “Comments to the Author” section, enter your conflict of interest statement in the “Confidential to Editor” section, and submit your "Accept" recommendation.

Reviewer #3: (No Response)

Reviewer #4: All comments have been addressed

2. Is the manuscript technically sound, and do the data support the conclusions?

Reviewer #3: Yes

Reviewer #4: Yes

3. Has the statistical analysis been performed appropriately and rigorously? 

Reviewer #3: Yes

Reviewer #4: Yes

4. Have the authors made all data underlying the findings in their manuscript fully available?

Reviewer #3: Yes

Reviewer #4: Yes

5. Is the manuscript presented in an intelligible fashion and written in standard English?

Reviewer #3: Yes

Reviewer #4: No

6. Review Comments to the Author

Reviewer #3: Major:

- The writing does not adhere to conventional scientific writing format making it difficult and unpleasant to read. I would suggest structuring the article as introduction, methods, results, discussion. There are also many subheadings that do not really have a core message. Additionally, the article consists of 11 figures each with only 1 panel. Can you condense this into less figures with more panels on each figure?

- Can you describe more about the LUNA16 dataset? In particular, what makes this dataset the gold standard for malignant vs benign pulmonary nodule discrimination? This is crucial for the study as it is what you are using for benchmarking

Minor:

- Table 3 should be presented as a proper figure with graphs. Benchmarking your method against other existing methods is a crucial part of the paper and it should be highlighted accordingly

- Can you also perform leave-one-out cross validation in addition to 5 fold cross validation?

- Lengthy descriptions of what basic statistical methods you used are unnecessary e.g. five-fold cross validation, detracting from the core message of your research and further contribute to the unpleasant readability of the paper.

Reviewer #4: The author presents a good manuscript.However , I feel the article can be improved by the use a of professional English language editor. for example the last sentence of line 47 and 48 is not grammatically correct.

Secondly would the addition of sensitivity, specifity, positive predictive value and the negative predictive value provide better results for comparison of the two methods better than as currently presented? Would a Kappa test be useful?

7. PLOS authors have the option to publish the peer review history of their article (what does this mean?). If published, this will include your full peer review and any attached files.

Reviewer #3: No

Reviewer #4: No

---

## [Author Response · Author response to Decision Letter 2]

18 Jan 2025

Response to the comments

Dear Editor and Reviewer:

Thank you for your letter and the reviewer’s comments concerning our manuscript entitled Cross-ViT based benign and malignant classification of pulmonary nodules. Those comments are all valuable and very helpful for revising and improving our paper, we have studied comments carefully and made corrections which we hope to meet with approval. We changed the structure of the article and tweaked the order and structure of some graphs. The responses to the reviewers’ comments are as follows.

Reviewer #3: Major:

- The writing does not adhere to conventional scientific writing format making it difficult and unpleasant to read. I would suggest structuring the article as introduction, methods, results, discussion. There are also many subheadings that do not really have a core message. Additionally, the article consists of 11 figures each with only 1 panel. Can you condense this into less figures with more panels on each figure?

[Response]: Thank you so much for your suggestion. We merged Chapters 1 and 2 and placed the Dataset section at the beginning of Methods. In addition, we optimized some titles. We reduced the number of images by merging some images together. The rearranged sections include: 'Introduction', 'Material and methods', ' Results', and ' Conclusion'.

- Can you describe more about the LUNA16 dataset? In particular, what makes this dataset the gold standard for malignant vs benign pulmonary nodule discrimination? This is crucial for the study as it is what you are using for benchmarking

[Response]: Thank you so much for your suggestion. We modified the description of the LUNA16 dataset. The LUNA16 dataset stands for Lung Nodule Analysis 16. This dataset was introduced in 2016 to develop a CAD system that can automatically detect lung nodules in CT scans.

Minor:

- Table 3 should be presented as a proper figure with graphs. Benchmarking your method against other existing methods is a crucial part of the paper and it should be highlighted accordingly

[Response]: Thank you so much for your suggestion. We plotted Table 3 as a histogram and added Fig 9.

- Can you also perform leave-one-out cross validation in addition to 5 fold cross validation?

[Response]: Thank you so much for your suggestion. The processed data in this paper contains 1004 lung nodules. If leave-one-out cross validation is used, 1004 models need to be trained, which requires a lot of time for image-type data. We believe that 5-fold cross validation is good enough to validate the results of this paper, while leave-one-out cross validation would be difficult to implement.

- Lengthy descriptions of what basic statistical methods you used are unnecessary e.g. five-fold cross validation, detracting from the core message of your research and further contribute to the unpleasant readability of the paper.

[Response]: Thank you so much for your suggestion. We removed the description of some common methods in ‘Evaluation metrics’ and ‘5-Fold cross-validation’.

Reviewer #4: The author presents a good manuscript.However , I feel the article can be improved by the use a of professional English language editor. for example the last sentence of line 47 and 48 is not grammatically correct.

[Response]: Thank you so much for your suggestion. We modified the syntax where you pointed out and checked the syntax of the full paper.

Secondly would the addition of sensitivity, specifity, positive predictive value and the negative predictive value provide better results for comparison of the two methods better than as currently presented? Would a Kappa test be useful?

[Response]: Thank you so much for your suggestion. Sensitivity, specifity, positive predictive value and the negative predictive value are generally not used together with ACC,PRE and REC because these two groups of indicators have similar effects. We add the kappa index in Table II, and the results show good consistency, which also reflects the better recognition effect of the proposed method for positive samples.

---

## [Editor Report · Decision Letter 3]

21 Jan 2025

Cross-ViT based benign and malignant classification of pulmonary nodules

PONE-D-24-34082R3

Dear Dr. Zhu,

We’re pleased to inform you that your manuscript has been judged scientifically suitable for publication and will be formally accepted for publication once it meets all outstanding technical requirements.

Kind regards,

Mohammad Amin Fraiwan

Academic Editor

PLOS ONE
---

## [Editor Report · Acceptance letter]

25 Jan 2025

PONE-D-24-34082R3 

PLOS ONE

Dear Dr. Zhu, 

I'm pleased to inform you that your manuscript has been deemed suitable for publication in PLOS ONE. Congratulations! Your manuscript is now being handed over to our production team.

Kind regards, 

on behalf of

Dr. Mohammad Amin Fraiwan 

Academic Editor

PLOS ONE